# Comparing Two Different Development Methods of External Parameter Orthogonalization for Estimating Organic Carbon from Field-Moist Intact Soils by Reflectance Spectroscopy

**Wu Yu [1,2], Yongsheng Hong [2,3], Songchao Chen [4], Yiyun Chen [3,5] and Lianqing Zhou [1,2,*]**

1   College of Resource and Environment, Tibet Agricultural and Animal Husbandry University, Linzhi 860000, China; 12114085@zju.edu.cn
2   Institute of Agricultural Remote Sensing and Information Technology Application, College of Environmental and Resource Sciences, Zhejiang University, Hangzhou 310058, China; hys@whu.edu.cn
3   School of Resource and Environmental Sciences, Wuhan University, Wuhan 430079, China; chenyy@whu.edu.cn
4   ZJU-Hangzhou Global Scientific and Technological Innovation Center, Hangzhou 311200, China; chensongchao@zju.edu.cn
5   State Key Laboratory of Soil and Sustainable Agriculture, Chinese Academy of Sciences, Nanjing 210008, China
*   Correspondence: lianqing@zju.edu.cn

**Abstract:** Visible and near-infrared (Vis–NIR) spectroscopy can provide a rapid and inexpensive estimation for soil organic carbon (SOC). However, with respect to field in situ spectroscopy, external environmental factors likely degrade the model accuracy. Among these factors, moisture has the greatest effect on soil spectra. The external parameter orthogonalization (EPO) algorithm in combination with the Chinese soil spectroscopic database (Dataset A, 1566 samples) was investigated to eliminate the interference of the external parameters for SOC estimation. Two different methods of EPO development, namely, laboratory-rewetting archive soil samples and field-collecting actual moist samples, were compared to balance model performance and analytical cost. Memory-based learning (MBL), a local modeling technique, was introduced to compare with partial least square (PLS), a global modeling method. A total of 250 soil samples from Central China were collected. Of these samples, 120 dry ground samples (Dataset B) were rewetted to different moisture levels to develop EPO *P1* matrix. Seventy samples (Dataset C) containing field-moist intact and laboratory dry ground soils were used to establish EPO *P2* matrix. The remaining 60 samples (Dataset D) also containing field-moist intact and laboratory dry ground soils were employed to validate the spectral models developed based on Dataset A. Results showed that EPO could correct the effect of external factors on soil spectra. For PLS, the validation statistics were as follows: no correction, validation $R^2 = 0.02$; *P1* correction, validation $R^2 = 0.56$; and *P2* correction, validation $R^2 = 0.57$. For MBL, the validation results were as follows: no correction, validation $R^2 = 0.06$; *P1* correction, validation $R^2 = 0.65$; and *P2* correction, validation $R^2 = 0.69$. The *P2* consistently yielded better results than *P1* did but simultaneously increased the sampling time and economic cost. The use of the *P1* matrix and the MBL algorithm was recommended because it could reduce the cost of establishing in situ models for SOC.

**Keywords:** visible and near-infrared spectroscopy; soil organic carbon; soil moisture; external parameter orthogonalization; local modeling

## 1. Introduction

Soil organic carbon (SOC) plays an important role in reducing greenhouse gas emissions into the atmosphere [1–5]. It is also an important indicator of soil quality and can improve soil biological productivity and agricultural sustainability [6,7]. Therefore, SOC content should be accurately and timely assessed for managing, enhancing, and improving

the utilization of this resource. With advancements in proximal soil sensing, visible and near-infrared (Vis–NIR) spectroscopy has become an effective technique for enriching conventional soil surveys to reduce cost and quantify multiple soil attributes [8–12]. The Vis–NIR spectra contain comprehensive soil information, including color, particle size, organic matter, and clay mineral. In comparison with traditional methods for determining SOC, Vis–NIR spectroscopy has several advantages, such as time saving, cost effectiveness, and being environmentally friendly.

The benefits of Vis–NIR spectroscopy have contributed to the development of many large spectral libraries at different local, regional, national, and even global scales [12,13]. All of these databases contain information on soils that have been prepared under dry ground conditions. Their spectral data and target properties have been, respectively, recorded and measured in laboratories. Thus, they involve substantial financial investment and should be exploited fully in practice. However, in applying such databases to field-moist intact soils, field spectra are affected by some external factors (henceforth denoted as in situ factors), such as variable soil moisture, surface conditions, and temperature variations; consequently, field spectra differ from laboratory dry spectra [14–19]. Thus, the prediction error of field spectra likely increases when spectroscopic calibration models are developed from laboratory dry ground soils [20–22]. The effect of moisture on soil spectra is complex and nonlinear, which are the main reasons for unsatisfactory model performance.

To overcome the influence of unfavorable factors on field spectra, improve the prediction accuracy of models, and eliminate external parameters, researchers proposed and adopted many methods, including external parameter orthogonalization (EPO), direct standardization, moisture classification, removal of wavelengths affected by moisture, slope bias correction, orthogonal signal correction, and generalized least squares weighting [16,17,23–27]. Among these methods, the EPO algorithm is widely used because it is easily applied and understood. The EPO algorithm decomposes the field spectral data into two parts: the useful part for the target property and the useless part affected by the external parameters. The calibrated spectral model is insensitive to external effects and more accurate when it is employed to predict field-moist soils by removing the in situ factors from spectra through EPO. EPO correction does not require prior knowledge of information on soil moisture; thus, it can be used without collecting additional measurements [14]. The EPO transformation matrix can be obtained by either rewetting archive soil samples or collecting field actual moist samples. Both methods have unique advantages and disadvantages. Most studies have used the former to develop an EPO transformation matrix and then to predict the target properties of laboratory rewetting samples, but such studies have not verified the prediction accuracy of EPO on field moist intact soils [17–19,26,28]. Some studies have adopted the latter to develop EPO [14,15,20,21]. Although the latter can effectively account for other factors, except soil moisture, it requires high cost and considerable labor to collect field soil samples; thus, it is challenging. Therefore, to balance model accuracy and logistical requirements, researchers should carefully compare these two EPO development methods.

Mathematical models are needed to establish the relationship between spectral data and target property. In general, partial least square (PLS) is the mainstream modeling technique of EPO-corrected spectra [14,15,18,20,21]. However, the calibration dataset of the EPO algorithm is usually an existing spectral library, which collects samples from different geographical regions and contains complex and nonlinear data relationships [12,13,29,30]. Thus, the spectral variation associated with soil properties can be locally stable [31]. In modeling with a large spectral library, we should determine whether all soil samples in a library should be used or a small subset of a library should be selected for modeling [32,33]. The use of all samples in a large and complex spectral library results in large-scale and universal models. They include more variations in spectral data and soil properties and provide higher predictive uncertainties. On the contrary, a smaller subset leads to a small-scale model suitable for local samples, but it may have better prediction accuracy [34]. In local

regression, each validation sample is estimated with a different calibration equation. This type of modeling technique can effectively remove unrelated or uninformative samples [30]. For example, memory-based learning (MBL) aims to use the most similar samples selected from the calibration dataset to predict each new sample [31]. The samples used to train a local model are chosen from a spectral database on the basis of their similarity to the predicted samples. However, when correction is performed for in situ field spectral data, the MBL algorithm has yet to be used to predict SOC via Vis–NIR spectra.

This study aimed to (1) investigate the effects of moisture and other in situ factors on reflectance spectra, (2) compare the model performance of PLS and MBL methods established without and with EPO correction on moist soils and explore the influence of EPO correction on important wavelengths in SOC estimation, and (3) compare the EPO transformation matrices developed from laboratory-rewetted archive soil samples and field-collected actual moist samples.

## 2. Materials and Methods

### 2.1. Soil Datasets

The soil data used in this study was divided into four independent datasets:

Dataset A: This subset consisted of 1566 topsoil (0–20 cm) samples gathered from 14 provinces in China. It contained 16 soil groups, including Anthrosols, Phaezems, Chemozems, Eutric Cambisols, and Luvisols [35]. All of the soil samples were air dried, ground, and sieved (2 mm mesh). Reflectance spectra were obtained under laboratory conditions with an ASD Fieldspec Vis–NIR spectrometer (Analytical Spectral Devices, Boulder, CO, USA) in a range of 350–2500 nm with a spectral resolution of 3 and 10 nm for the regions of 350–1000 and 1000–2500 nm, respectively. A standardized white panel was applied to calibrate this spectrometer. For each soil sample, 10 spectra were scanned and then averaged to one spectrum. The SOC content was chemically measured through $H_2SO_4$-$K_2Cr_2O_7$ oxidation [36]. Further details about this soil dataset were presented by Ji et al. [29] and Shi et al. [37]. This set was designed to develop a dry-ground calibration model.

Dataset B: This set comprised 120 dry ground samples collected from Honghu City, Hubei Province (Figure 1), and this study area is located in the Jianghan Plain. Its terrain is wide and flat, its altitude is mostly between 22 and 29 m, and its geomorphologic type is an alluvial plain primarily developed from Quaternary alluvium and lake sediments. The area is within a subtropical humid monsoon climate zone with mean annual temperature and precipitation of 16.4 °C and 1195.8 mm, respectively. The prevailing soil types of these 120 samples are Anthrosols and Cambisols [35], and the main crops are rice, cotton, and rapeseed. All of the sampling points were collected in June 2013 and geolocated with a handheld global positioning system. The dry ground samples in this group were wetted and air dried to different soil moisture levels. At each moisture level, the reflectance spectra were scanned with the ASD spectrometer. For each sample at every moisture level, an average of 10 reflectance spectra produced one reflectance spectrum. The procedure for sample rewetting was elaborated in the next section. The SOC content was determined via the potassium dichromate method [36]. The rewetted spectra in this group were used to develop the EPO projection matrix *P1*.

Dataset C and Dataset D: These two sets consisted of 70 and 60 samples, respectively, which were also collected over Jianghan Plain, Hubei Province (Figure 1). The topographic characteristics, geomorphologic types, climatic conditions (i.e., temperature and precipitation), soil types, and crop types were similar to those in Dataset B. The spectral data collected under field conditions generally contain unfavorable noises because of the effect of solar radiation, atmosphere, and water factors. Herein, field-moist intact samples were transported to the laboratory, and their spectra were obtained in the laboratory with the ASD spectrometer. At each sampling point in the field, the soil water content was recorded with FieldScout TDR 300 (Spectrum Technologies Inc., Aurora, IL, USA). The samples were then air dried, milled and sieved to pass a 2 mm mesh from which laboratory dry ground

spectra were obtained. Each reflectance spectrum was obtained as the mean value of 10 spectra. Both sets (i.e., Dataset C and Dataset D) contained pairs of Vis–NIR spectra from field-moist intact and laboratory dry ground soils. Similar to the SOC content in Dataset B, the SOC content in Dataset C and Dataset D was also measured via the potassium dichromate method [36]. The field-moist intact and laboratory dry ground spectra in Dataset C were used to develop the EPO projection matrix *P2*. Dataset D was used to test the effectiveness of EPO in removing the effects of soil moisture and other in situ factors on SOC estimation.

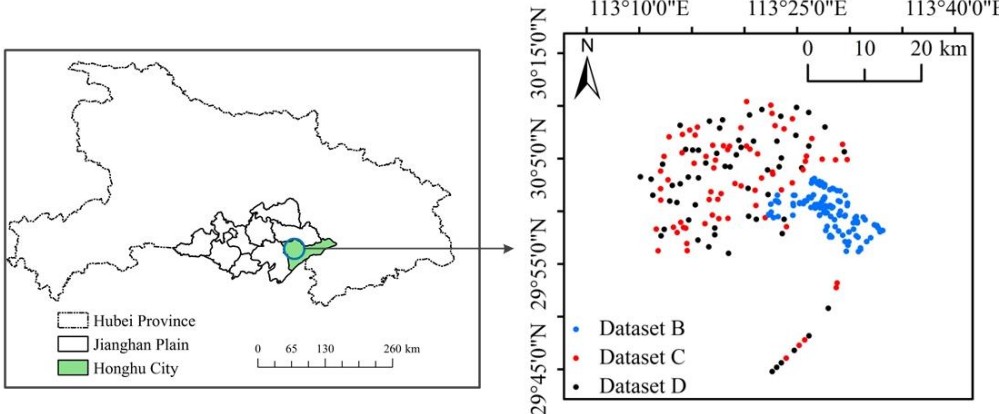

**Figure 1.** Study area showing the sampling points of Dataset B, Dataset C, and Dataset D.

## 2.2. Rewetting Experiment

Subsamples (approximately 100 g) were extracted from the original dry ground soils in Dataset B. Before the rewetting experiment was conducted, all of these 120 subsamples were oven dried at 105 °C for 1 day to eliminate soil moisture. Sample rewetting was completed in only one batch. A spray flask was used to slowly add water to the soil. The subsamples were placed in Petri dishes, and approximately 40 g of deionized water was used to wet each sample to obtain a moisture level of approximately 40% (dry basis, gravimetric). This step was immediately followed by covering the dishes with lids for 24 h to avoid moisture loss and homogenize the moisture within the sample. The samples were weighed with a scale to determine the soil moisture content and record their first set of wet spectra. On the following days, the samples were uncovered, air dried at room temperature, and weighed daily. Simultaneously, their corresponding wet spectra were recorded. This process was repeated until the soil moisture was close to 0%. Consequently, eight soil moisture levels were obtained. All of the moisture contents were gravimetric; thus, they were calculated on the basis of dry soils.

## 2.3. Spectral Preprocessing

The final output interval of the spectrometer was 1 nm, thereby providing us 2151 reflectance values for each spectrum. The original spectral curves were smoothened through Savitzky–Golay filtering with a window size of 11 nm and a second-order polynomial [38]. Spectral regions ranging from 350 nm to 399 nm and from 2401 nm to 2500 nm were excluded from subsequent data analysis because of spectral noise. Spectra were resampled to a 10 nm resolution to reduce the dimension of spectral matrix and computation time; thus, 201 wavelength variables were obtained. Continuum removal (CR) allows the normalization of the reflectance spectra to isolate and identify significant absorption characteristics [39]. Savitzky–Golay smoothing and CR transformation were processed in R programming language with the *prospectr* package [40,41].

## 2.4. EPO

The EPO algorithm introduced by Roger et al. [42] aims to remove the effects of external environmental parameters on reflectance spectra. EPO finds spectral areas affected by soil

moisture and then projects the reflectance spectra into a space orthogonal to variation [17]. Field-moist intact spectra (**X**) can be expressed as a combination of the useful part (**XP**, containing useful spectral information of SOC), the external part (**XQ**, including useless information affected by water content and other in situ factors), and the spectral noise (**R**). EPO aims to isolate useful spectral response by developing the projection matrix **P**. **D** is initially defined as the spectral difference between dry and moist (laboratory rewetted or field-moist intact) soils. In summary, the EPO method proceeds as follows:

(1)　Calculate the difference matrix **D**.
(2)　Perform singular value decomposition on matrix **D** to obtain matrix **V**. This process can also be achieved by employing principal component decomposition on $\boldsymbol{D}^{\mathrm{T}}\boldsymbol{D}$. The superscript T represents the matrix transpose.
(3)　Define the dimension g of the EPO and calculate a subset $\boldsymbol{V}_{\mathrm{s}}$ of the **V** matrix.
(4)　Calculate **Q** from $\boldsymbol{V}_{\mathrm{s}}\boldsymbol{V}_{\mathrm{s}}^{\mathrm{T}}$.
(5)　Derive the projection matrix **P** from **I**–**Q**, where I is an identity matrix.

EPO data analysis was conducted in the R statistical environment [40]. During EPO development, the number of dimensions $g$, which is an important parameter, should be determined in the EPO. $g$ was calculated on the basis of Wilk's Λ method [42,43] defined as follows:

$$\Lambda \ = \ \frac{trace(\boldsymbol{B})}{trace(\boldsymbol{T})}, \tag{1}$$

where **T** is the total covariance matrix of the EPO-transformed reflectance spectra, and **B** is the inter-group covariance matrix of the EPO-transformed reflectance spectra (i.e., averaging over different moisture levels for each sample). The reflectance spectra transformed by EPO are projected in a different subspace; thus, in the development of the prediction models, calibration and validation datasets should be recalibrated with the EPO projection matrix.

### 2.5. Statistical Analysis and Modeling

The SOC data was investigated by descriptive statistics, including minimum, maximum, and mean values and coefficient of variation (CV). Principal component analysis (PCA) was applied to evaluate the dispersion of the reflectance spectra and to determine the spectral similarities or differences among various datasets [44]. PCA scores were used to interpret these features. The remaining principal components (PCs) represent a small percentage of the spectral information, so they were not used later [12]. Two multivariate methods, namely, PLS and MBL, were used to build SOC prediction models.

#### 2.5.1. PLS

The PLS can effectively deal with data multi-collinearity and cases in which the number of spectral variables exceeds the number of samples [45–47]. It can be considered as a combination of correlation analysis, PCA, and regression analysis. PLS is a linear method based on the projection of predictive variables and response variables in a group of latent factors and corresponding scores, and it is the most commonly used model for predicting soil properties from spectral data [18,48,49]. Unlike some nonlinear methods, PLS models are less computationally intensive and more interpretable [32]. The determination of important wavelengths in the PLS model can be obtained on the basis of the variable importance in projection (VIP) [50]. If a VIP score at a specific wavelength is greater than 1, then it indicates that this wavelength is necessary to estimate SOC [51]. In the present research, the optimal number of latent variables was selected through leave-one-out cross-validation, and PLS was implemented in R with the *pls* package [52].

#### 2.5.2. MBL

In machine learning theory, MBL is a data-driven method similar to human reasoning, which includes remembering past situations, adjusting them to solve current problems,

examining the possibility of using new approaches to solve problems, and remembering experiences to improve knowledge [31]. MBL resembles the k-nearest neighbor regression, locally weighted PLS regression, and LOCAL algorithm. MBL does not generate a global function; instead, it relies on a reference set or a spectral library to perform local regression. MBL mainly comprises three steps:

(1) Nearest-neighbor searching. This step aims to determine which samples in the calibration dataset are similar to the samples in the validation dataset, that is, either similar or dissimilar measurements are needed. The optimized PC Mahalanobis (oPC-M) distance is implemented to indicate similarity or dissimilarity [53]. Selecting the optimal number of PCs is based on the minimal root mean square of compositional differences.

(2) Training and testing. This step is performed in the spectral space. For each sample to be predicted, its most similar sample, namely, its k-nearest neighbor, must be used to fit the model. However, before fitting is performed, a sufficient number of neighbors must be identified for each calibration. A Gaussian process regression with a linear covariance function (GPL) is applied to predict samples in the validation dataset.

(3) Fitting and predicting. A new local GPL model is developed for each sample in the validation dataset. The predictors include spectral data and a local distance matrix.

In the MBL method, the following parameters should be optimized: the optimal number of PCs for oPC-M distance calculation and the number of the most similar samples used for local GPL regression. In this study, the maximum number of PCs was set to 50, and 23 different values of the most similar samples ranging from 120 to 810 were tested with an increment of 30 samples at each step. Further details on the descriptions of the MBL algorithm were provided by Ramirez-Lopez et al. [53] and Ramirez-Lopez et al. [31]. For MBL modeling, package *resemble* was used in the statistical software R [43].

The model performance was assessed using the following indices: the coefficient of determination ($R^2$), root-mean-square error (*RMSE*) (Equation (2)), the ratio of performance to inter-quartile range (*RPIQ*) (Equation (3)), and *bias* (Equation (4)).

$$RMSE = \sqrt{\frac{\sum(y_{pre} - y_{meas})^2}{n}} \tag{2}$$

$$RPIQ = \frac{IQ}{RMSE} \tag{3}$$

$$bias = \frac{\sum(y_{pre} - y_{meas})}{n} \tag{4}$$

where $y_{pre}$ and $y_{meas}$ are the predicted and measured values of SOC, $n$ is the number of samples, and IQ is the interquartile range of the measured SOC. A well-behaved model typically has large $R^2$ and RPIQ and small RMSE and bias.

### 2.6. Modeling Flowchart

Model calibration and validation are shown in Figure 2. Here, we mainly explored four types of models to study the effects of EPO algorithms for correcting moisture and other in situ factors on the spectral estimation of SOC. Both PLS and MBL models were involved in modeling.

Prediction I (Figure 2a): Dataset A (dry ground soils) was applied to build estimation models, which were then tested on the dry ground spectra in Dataset D. This prediction was used as a benchmark to compare with the three other predictions.

Prediction II (Figure 2b): Dataset A (dry ground soils) was used to establish estimation models, which were used to predict the field-moist intact spectra in Dataset D. In this process, no effort was made to remove the effect of moisture and other in situ factors.

Prediction III (Figure 2c): Dataset B with rewetted spectra at different moisture levels was used to develop the EPO projection matrix *P1*. The dry ground spectra in Dataset A

were subsequently transformed with **P1**, and the transformed Dataset A* was applied to calibrate an EPO model. The matrix **P1** was also employed to the field-moist intact spectra in Dataset D. The EPO model was used to predict the EPO-transformed moist intact spectra in Dataset D*.

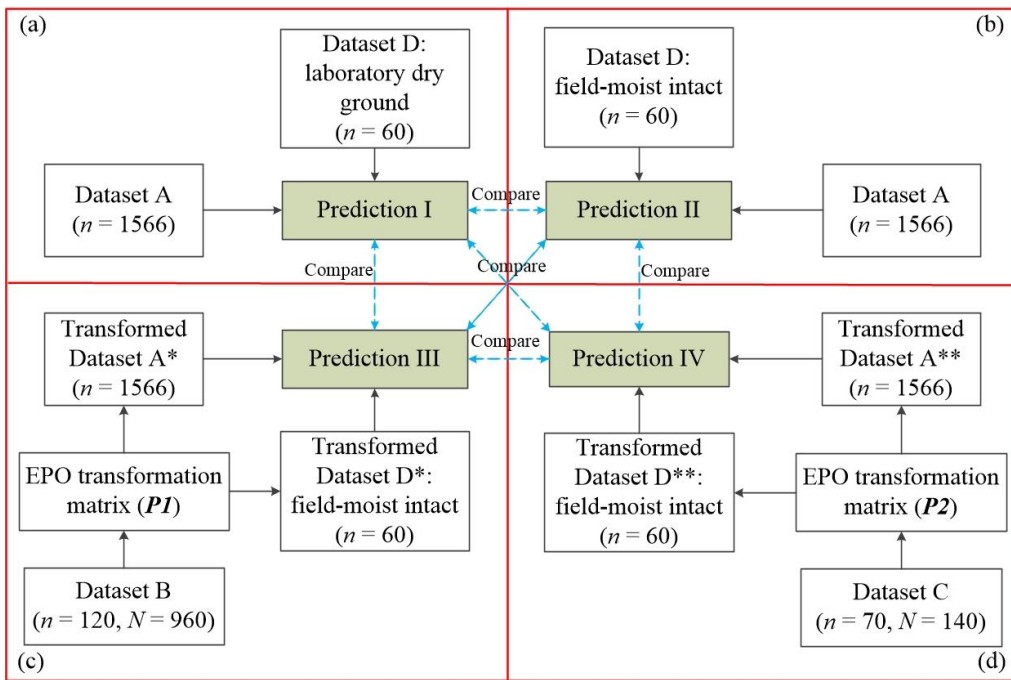

**Figure 2.** Flowcharts of the four approaches used to estimate SOC: (**a**) Dataset A model applied to estimate laboratory dry ground samples. (**b**) Dataset A model applied to estimate field-moist intact samples. (**c**) Dataset A model first recalibrated with **P1** correction and then applied to estimate **P1** transformed field-moist intact samples. (**d**) Dataset A model was initially recalibrated with **P2** correction and then applied to estimate **P2** transformed field-moist intact samples.

Prediction IV (Figure 2d): Dataset C comprising laboratory dry ground and field-moist intact spectra was utilized to obtain the EPO projection matrix **P2** for transforming Dataset A and field-moist intact spectra in Dataset D. A moisture-insensitive EPO model based on the transformed Dataset A** was calibrated. This model was tested on the EPO-transformed moist intact spectra in Dataset D**.

The prediction results of these four types of models were compared with one another to understand the performance of **P2** compared with that of **P1** in accounting for moisture and other in situ factors and to compare the performance of the PLS model with that of the MBL model.

## 3. Results and Discussion

### 3.1. Soil Moisture Content and SOC

In Dataset B, the average soil moisture contents at the eight moisture levels were 32.61%, 29.05%, 25.45%, 21.58%, 16.91%, 11.83%, 6.87%, and 2.56% (Figure 3). In Dataset C, for the field-moist intact samples, the soil moisture content varied from 11.43% to 36.48%, with a mean value of 22.51% (Figure 3). Soil moisture of field-moist intact samples in Dataset D ranged from 12.44% to 39%, which was the same as that in Dataset C. Overall, the rewetting experiment obtained a large range of soil moisture contents from 0% to 40.71%, completely covering a range of soil moisture contents of the field-moist intact samples in Dataset D.

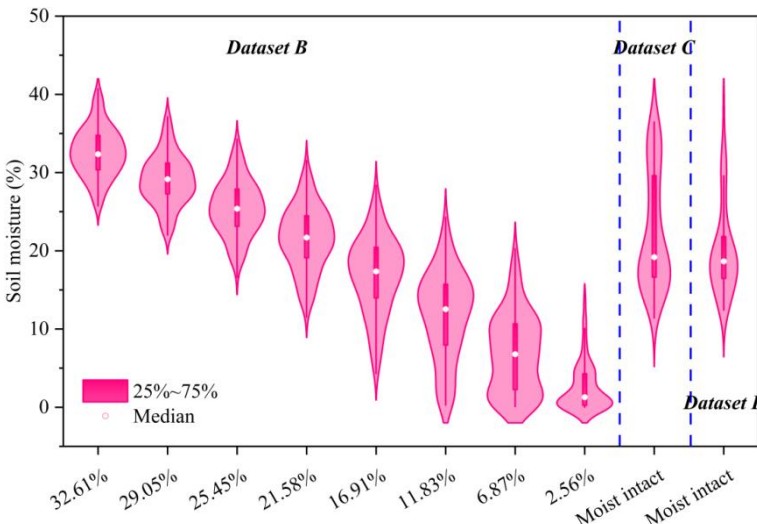

**Figure 3.** Soil moisture values measured during the rewetting experiment (Dataset B) and under the field-moist intact conditions (Dataset C and Dataset D).

The descriptive analysis of the SOC values in four different datasets was presented in Table 1. The range of the SOC values in Dataset A was quite large, that is, from 0.10% to 4.34%. Wilding [54] divided the CV values into three groups: CV < 15%, low variability; 15% < CV < 35%, moderate variability; and CV > 35%, high variability. The CV of Dataset A was high, whereas the CVs of Dataset B, Dataset C, and Dataset D were moderate as a possible result of differences in soil type, land use, parent material, and spatial extent [12,55,56]. In comparison with Dataset B, Dataset C, or Dataset D, Dataset A contained a larger range in SOC with lower minimum and higher maximum SOC values. These results indicated that Dataset A contained soil samples that were not represented in Dataset B, Dataset C, or Dataset D. Thus, the estimation model provided a favorable predictive performance.

**Table 1.** Summary statistics of soil organic carbon (SOC) in different subsets.

| Sample Sets | $n$ [1] | N [2] | Min (%) | Max (%) | Mean (%) | Standard Deviation (%) | CV (%) [3] |
|---|---|---|---|---|---|---|---|
| Dataset A | 1566 | 1566 | 0.10 | 4.34 | 1.35 | 0.67 | 49.61 |
| Dataset B | 120 | 960 | 0.52 | 2.68 | 1.29 | 0.49 | 37.75 |
| Dataset C | 70 | 140 | 0.64 | 3.23 | 1.77 | 0.46 | 26.02 |
| Dataset D | 60 | 120 | 1.01 | 2.28 | 1.65 | 0.30 | 17.90 |

[1] Sample number. [2] Number of scans. [3] Coefficient of variation.

### 3.2. Effects of Moisture and Other In Situ Factors on Reflectance Spectra

In Dataset B with different soil moisture levels (Figure 4a), increasing the soil moisture content led to a decreased reflectance spectra, but the shift was not uniform along the wavebands. This feature is consistent with previously reported results [15,19,25,57,58]. However, the moisture-induced decrease in the reflectance spectra was less pronounced when the soil moisture was larger than 16.91%. This pattern can be attributed to the results of Lobell and Asner [59], who showed that additional soil moisture that continuously fills micro- and macropores slightly influences the reflectance spectra as soil moisture increases and when enough soil moisture content is absorbed by most of the soil surfaces. After CR transformation was completed (Figure 4b), three absorption features located within 400–580, 1340–1680, and 1840–2150 nm were noted. The difference in the region of 400–580 nm is presumably due to the interactions between soil moisture and soil color [60], whereas the differences in the regions of 1340–1680 and 1840–2150 nm can be related to water absorption areas and other OH bands [15,16,46,61]. The effect of soil moisture was stronger

in the NIR range (700–2400 nm) than in the visible range (400–700 nm), indicating that longer wavelengths are more appropriate for estimating soil moisture content.

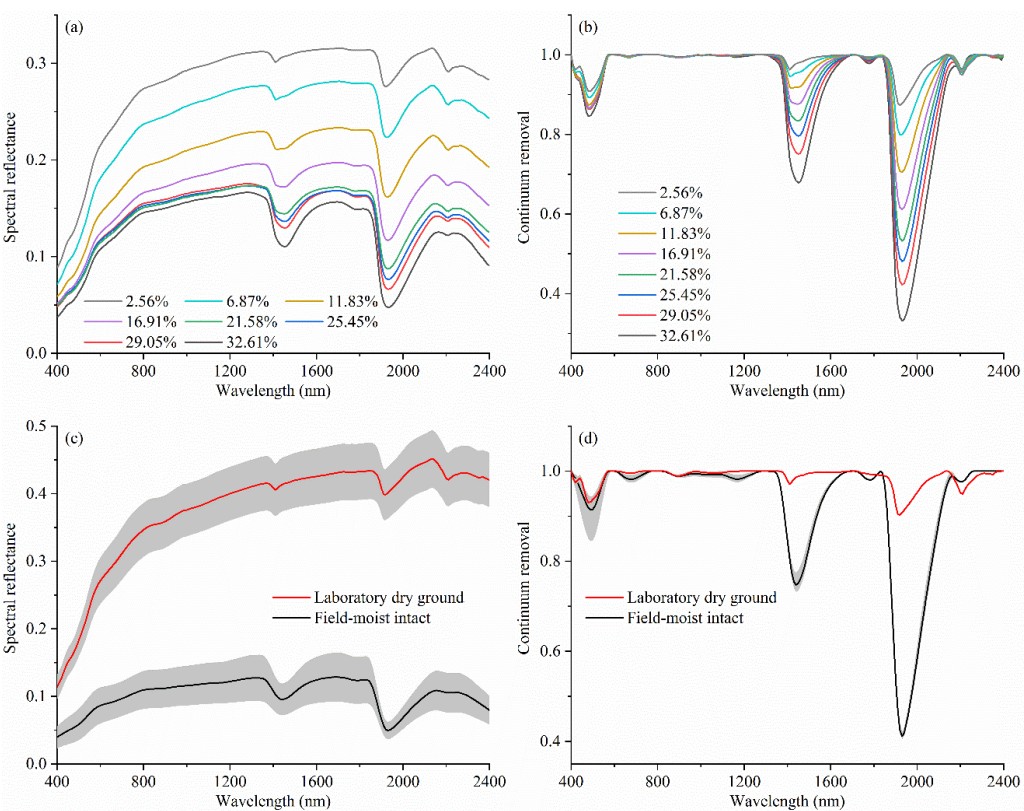

**Figure 4.** Average spectral and continuum removal (CR) reflectance: (**a**) mean reflectance spectra at different moisture levels (Dataset B, *n* = 120); (**b**) mean CR spectra at different moisture levels (Dataset B, *n* = 120); (**c**) mean reflectance spectra collected under laboratory dry ground and field-moist intact conditions (Dataset C, *n* = 70); and (**d**) mean CR spectra collected under laboratory dry ground and field-moist intact conditions (Dataset C, *n* = 70). The shaded areas in the lower subplots denote spectral standard deviations.

The average spectra of 70 samples from Dataset C measured under laboratory dry ground and field-moist intact conditions are shown in Figure 4c. The laboratory spectra were higher than the field spectra, which might be affected by various factors, such as soil moisture, roughness, and soil surface condition. After the spectral transformation was achieved by CR (Figure 4d), similar to the pattern in Figure 4b, three absorption features at 400–580, 1340–1680, and 1840–2150 nm were observed. The field spectra absorbed more light and had larger depths and widths at the water absorption bands around 1450 and 1940 nm than the laboratory spectra did.

The PCA was performed on the spectral data from Dataset B (Figure 5a), Dataset A, and Dataset C (Figure 5b). For Dataset B (Figure 5a), PC1 and PC2 explained more than 99% of the spectral variability. As soil moisture increased, the PC scatter points varied from right to left. The samples with moisture content ranging from dry ground to 11.83% could be easily discriminated, whereas the PC scores were mixed from 16.91% to 32.61% and could not be clearly distinguished. This phenomenon is consistent with that in Figure 4a. Adding water changes the physical structure and the color of the soils, further explaining the low sensitivity of spectral response to high moisture content [25,62]. For Dataset A and Dataset C (Figure 5b), the first two PCs accounted for more than 95% of the spectral variability. The first two PCs of the dry ground spectra in Dataset C overlapped with those of the spectra in Dataset A, thereby showing the spectral similarity between these two sets of spectra. However, overlapping was almost not observed between the moist intact

spectra in Dataset C and Dataset A because of the influence of the field external factors on the reflectance spectra, and the moist intact spectra in Dataset C were distributed in a separate feature space from the Dataset A spectra.

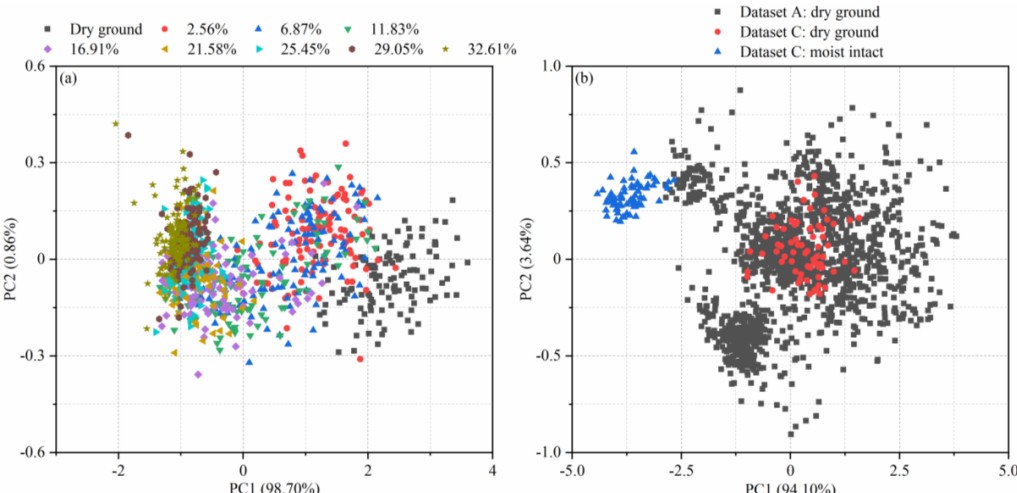

**Figure 5.** Principal component analysis (PCA): (**a**) score plot of reflectance spectra at different moisture levels (Dataset B) and (**b**) score plots of Dataset A and Dataset C (two different moisture conditions).

The above analyses can be summarized as follows. First, the bands of water absorption in the regions of 1340–1680 and 1840–2150 nm masked the sensitive spectral wavebands related to SOC prediction. Some previously published studies, such as those of Hong et al. [48], Jiang et al. [63], Nawar and Mouazen [64], and Viscarra Rossel and Behrens [46], showed that the wavelengths around 1400 and 1900 nm are spectrally important for SOC estimation. Second, the spectral feature space of field-moist samples did not overlap with that of Dataset A, often leading to an inaccurate SOC estimation. Therefore, a method for removing or minimizing the influence of external parameters on reflectance spectra is needed so that the existing spectral library of Dataset A can be used to predict field-moist samples.

### 3.3. Effects of Moisture and Other In Situ Factors on Spectral Estimation of SOC

Moisture correction using the EPO method was dependent on $g$, which was the number of EPO dimensions retained for calculating the transformation matrix $P$. Figure 6 showed Wilk's $\Lambda$ values as a function of $g$ for $P1$ and $P2$ development. The numbers of EPO components determined by Wilk's $\Lambda$ metrics for $P1$ and $P2$ were 5 and 4, respectively. In general, lower EPO dimensions provide considerably smoother corrected spectra than higher EPO dimensions do, that is, the latter obtain noisy spectra. The primary reason behind this pattern is that high EPO components may introduce additional spectral noise to the matrix $Q$ by including considerably numerous eigenvectors from the decomposition on matrix $D$. Wilk's $\Lambda$ method can determine the dimension of EPO only by spectral information, and it is simple and feasible in practical applications [19,43].

After projection with EPO was performed, the scores of the first two PCs derived through PCA in Dataset D and Dataset A were shown in Figure 7. Contrary to the observations in PCA conducted before EPO was applied (Figure 7a,b), the convex hulls of the moist spectra in Dataset D were contained within those of Dataset A, suggesting that the moist spectra in Dataset D occupied the same spectral space with Dataset A. Thus, after EPO was applied, the estimation models that were calibrated on Dataset A with dry soils could be expected to work on the field-moist intact soils.

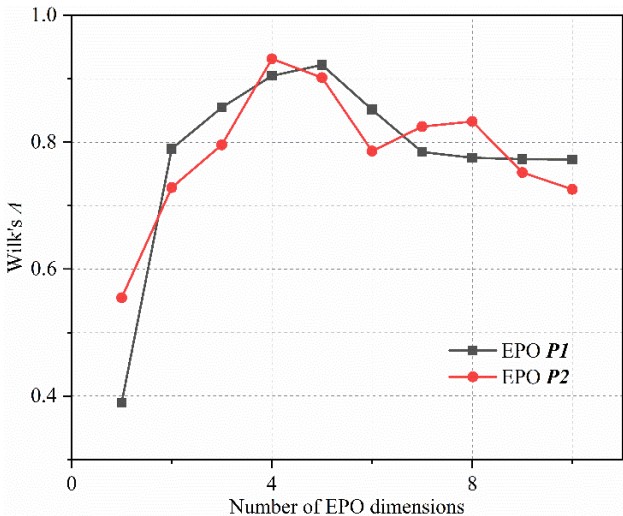

**Figure 6.** Selection of the optimum number of EPO dimensions (*g*). *P1* is developed by subtracting the rewetted reflectance spectra at different moisture levels from the laboratory dry ground reflectance spectra, and *P2* is developed by subtracting the reflectance spectra of field-moist intact samples from the corresponding reflectance spectra of laboratory dry ground samples.

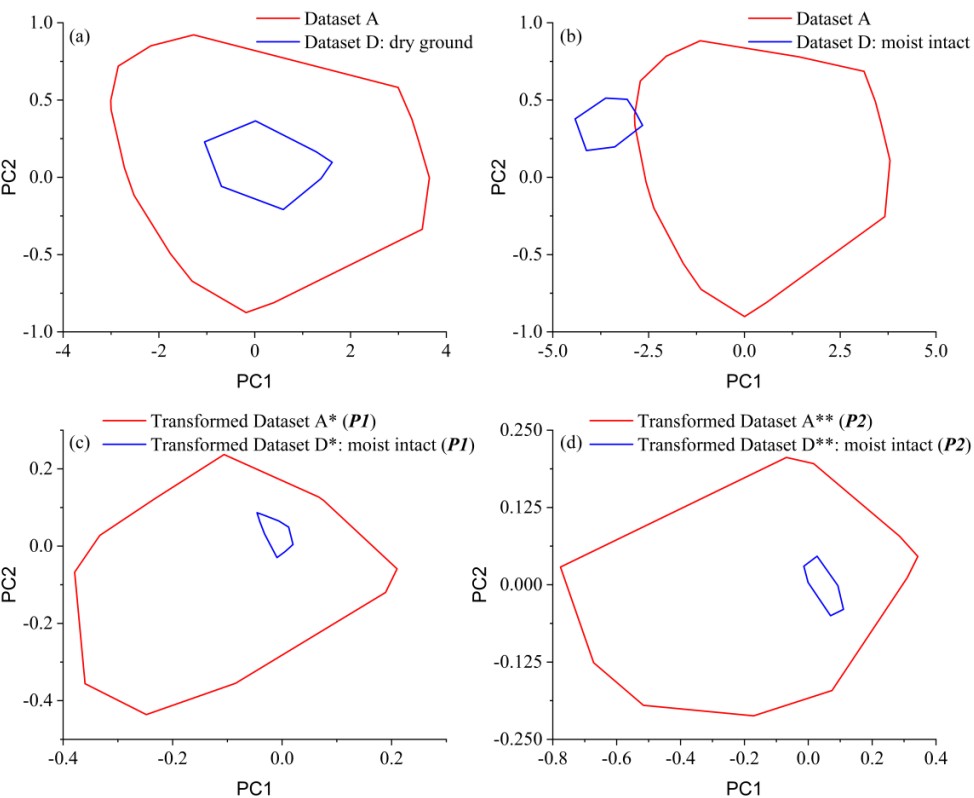

**Figure 7.** Score plots for the first and second principal components: (**a**) Dataset A and dry ground samples in Dataset D. (**b**) Dataset A and field-moist intact samples in Dataset D. (**c**) Dataset A* and field-moist intact Dataset C* corrected by *P1*. (**d**) Dataset A** and field-moist intact Dataset C** corrected by *P2*. Lines indicate the convex hulls of each dataset.

The performance of PLS and MBL models was tested to predict the SOC on dry ground and moist intact spectra before and after EPO transformation was carried out (Table 2). For the dry ground spectra, PLS and MBL methods provided good validation results for SOC, and PLS and MBL had RPIQ of 2.70 and 2.91, respectively. Conversely, without EPO application, the prediction performance of the PLS and MBL models was poor when

the moist intact spectra were directly used (PLS: $R^2$ = 0.02, RPIQ = 0.43; MBL: $R^2$ = 0.06, RPIQ = 1.25). This degradation in the model performance under the influence of moisture and other in situ factors is consistent with the results reported by other authors [14–17,20,21] and further confirms the rationality of efforts to develop a moisture correction algorithm. After the moist spectral transformation was conducted with the EPO algorithms, the model accuracies of PLS and MBL were greatly improved. Specifically, for EPO P1, PLS and MBL models had RPIQ of 2.27 and 2.53, respectively. For EPO P2, PLS and MBL models had RPIQ of 2.31 and 2.72, respectively. In principle, the EPO algorithm is a process of information removal to effectively remove the spectral information related to moisture and other factors, suggesting that spectral transformation with EPO can improve model performance [17,21,26]. However, overall, EPO predictions with PLS and MBL models for moist intact spectra (Table 2) were not as accurate as their corresponding PLS and MBL models for the dry ground spectra of the same soils. This result is expected, considering that laboratory dry ground spectra were scanned on dry ground soils; thus, they were not subjected to in situ effects.

**Table 2.** Validation results of partial least square (PLS) and memory-based learning (MBL) models for estimating SOC without and with external parameter orthogonalization (EPO) correction.

| Modeling Technique [1] | Calibration Dataset | Validation Dataset | Moisture Correction Approach [2] | N [3] | $R^2$ [4] | RMSE (%) [5] | RPIQ [6] | Bias (%) [7] |
|---|---|---|---|---|---|---|---|---|
| PLS | Dataset A | Dataset D (dry ground) | – | 26 | 0.69 | 0.1638 | 2.70 | 0 |
| | Dataset A | Dataset D (moist intact) | No correction | 31 | 0.02 | 1.0268 | 0.43 | 0.8884 |
| | Transformed Dataset A* | Transformed Dataset D* (moist intact) | EPO (*P1*) | 33 | 0.56 | 0.1951 | 2.27 | 0 |
| | Transformed Dataset A** | Transformed Dataset D** (moist intact) | EPO (*P2*) | 30 | 0.57 | 0.1918 | 2.31 | 0 |
| MBL | Dataset A | Dataset D (dry ground) | – | 540 | 0.73 | 0.1519 | 2.91 | 0.0047 |
| | Dataset A | Dataset D (moist intact) | No correction | 180 | 0.06 | 0.3538 | 1.25 | −0.0794 |
| | Transformed Dataset A* | Transformed Dataset D* (moist intact) | EPO (*P1*) | 600 | 0.65 | 0.1750 | 2.53 | −0.0040 |
| | Transformed Dataset A** | Transformed Dataset D** (moist intact) | EPO (*P2*) | 660 | 0.69 | 0.1628 | 2.72 | 0 |

[1] PLS and MBL mean partial least square and memory-based learning, respectively. [2] *P1* is developed by subtracting the rewetted reflectance spectra at different moisture levels from the laboratory dry ground reflectance spectra, whereas *P2* is developed by subtracting the reflectance spectra of field-moist intact samples from the corresponding reflectance spectra of laboratory dry ground samples. [3] Number of optimal latent variables in PLS models or number of the most similar samples in MBL models. [4] Determination coefficient. [5] Root-mean-square error. [6] Ratio of performance to interquartile range. [7] Average difference between measured and predicted values.

The MBL models (RPIQ, 1.25–2.91) outperformed the PLS models (RPIQ, 0.43–2.70) in all of the cases regardless of the dry ground spectra or the moist intact spectra transformed before and after EPO (Table 2). The MBL model typically aimed to choose the most similar samples from the spectral library to improve the prediction accuracy of each validation sample [31,65]. Thus, the MBL method can effectively discard potential outliers included in the spectral library because only some of the samples in the spectral library are useful for predicting SOC [32,33]. The MBL method yields better results when the collected samples likely contain complex and nonlinear relationships between SOC and spectral data, whereas the PLS method is a linear modeling technique suitable for linear relationships only [34,65,66]. Our model results support the findings of Jaconi et al. [67], who summarized that among different modeling strategies, calibration with the MBL algorithm provides the most accurate prediction for SOC. In MBL modeling techniques (Table 2), the number of the most similar samples in the moist intact spectra matched from Dataset A was much smaller than that in the dry ground spectra selected from Dataset A. The effects of in situ factors on field spectra made the determination of samples with similar spectra from Dataset A difficult. However, after EPO was performed to remove the in situ factors, the numbers of

spectrally similar samples increased to 600 and 660 for the EPO-transformed *P1* and *P2*, respectively.

Figure 8 illustrates how EPO improved the prediction accuracy of the moist samples. In predicting the laboratory dry spectra (Figure 8a), the prediction $R^2$ was 0.73, indicating a strong correlation between the measured and predicted SOC values. However, without EPO (Dataset A applied directly to moist intact spectra), the prediction was poor as indicated by the large RMSE value of 0.3538% (Figure 8b). Most of the sample points were either overestimated or underestimated, and the regression line deviated from the 1:1 line. For the moist spectra corrected by EPO algorithms (Figure 8c,d), EPO reduced RMSE values compared with the prediction of the moist intact spectra without correction, and most scatter points were distributed well around the 1:1 lines.

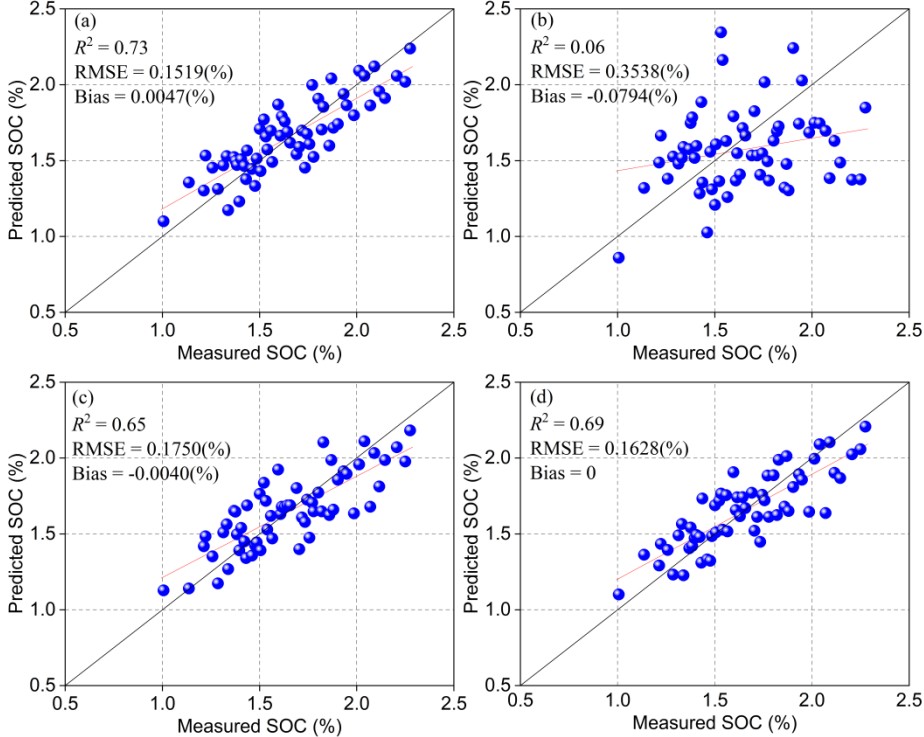

**Figure 8.** Scatterplots of SOC estimation from MBL: (**a**) prediction made on the spectra recorded in the laboratory (Dataset D: dry ground); (**b**) prediction made on the spectra recorded in the field (Dataset D: moist intact); (**c**) prediction made on the field spectra corrected by *P1* (transformed Dataset D*: moist intact); and (**d**) prediction made on the field spectra corrected by *P2* (transformed Dataset D**: moist intact). Black and red lines represent 1:1 and regression lines, correspondingly.

### 3.4. Comparison of P1 and P2 Transformation Matrices

Figure 9 visualizes *P1* and *P2* transformation matrices and three randomly selected spectral data before and after EPO transformation. The color bars of *P1* and *P2* are adjusted to a consistent range of values. Visually, *P1* and *P2* were significantly different (Figure 9c,d). Likewise, the transformed spectral data significantly varied after the EPO projection was completed (Figure 9e,f). With EPO correction, the transformed field moist spectral data were similar to each other.

Transforming the moist spectra through either *P1* or *P2* improved the prediction accuracy (Table 2). The prediction results of *P2* were better than those of *P1* (Table 2) regardless of PLS or MBL modeling techniques, mainly because the developed *P2* transformation matrix aimed to correct moisture and other in situ factors, such as aggregation and intactness, whereas *P1* was generated from the laboratory-rewetted samples presumably only suitable for correcting the moisture effect. This finding supports the result of Ge et al. [15] in correcting moisture and other in situ factors, that is, the prediction accuracy of the inclusion

of more representation of soil variability is higher than that of only inclusion of moisture variability.

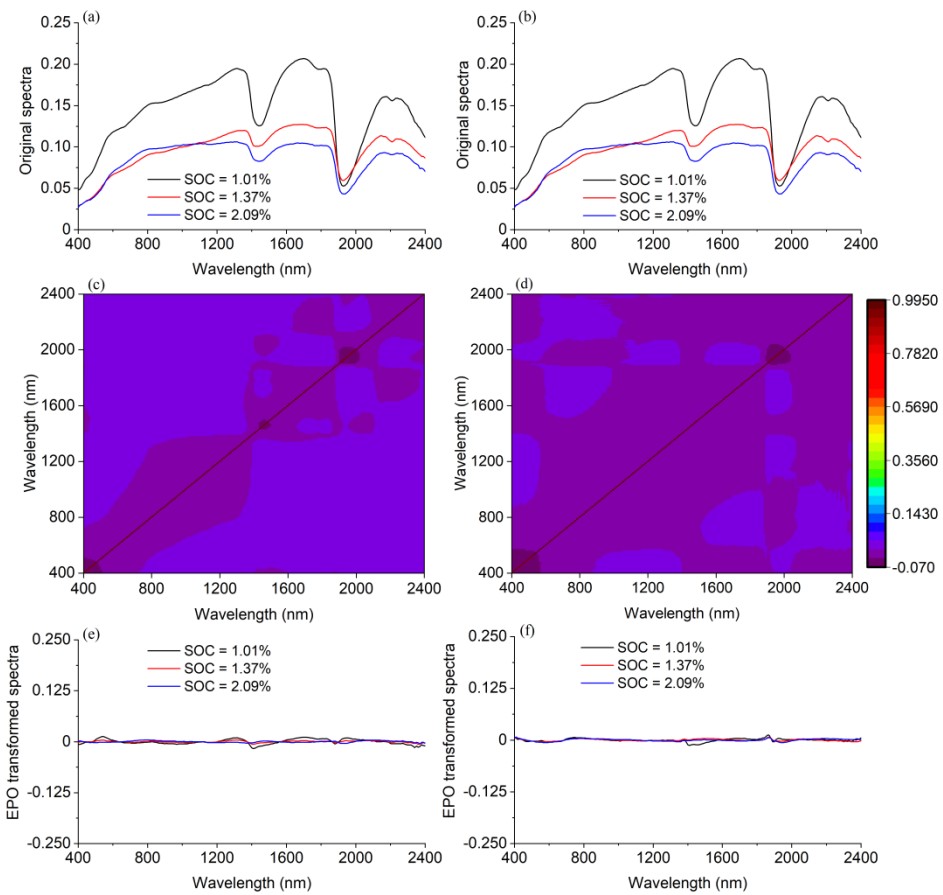

**Figure 9.** Visualization of EPO projection applied for spectral correction: (**a,b**) three randomly selected moist spectra from Dataset D; (**c**) a visual diagram of EPO *P1*; (**d**) a visual diagram of EPO *P2*; (**e**) spectral data transformed by *P1*; and (**f**) spectral data transformed by *P2*.

### 3.5. Important Wavelengths for SOC

We investigated whether the use of EPO affects the sensitive wavebands of SOC. Hence, we analyzed the VIP scores derived from different PLS models (Figure 10). Because MBL is a local model rather than a global model and cannot identify the important wavelengths for SOC. In general, EPO correction exerted no significant effect on the distribution of the important waveband of SOC (Figure 10). The important wavelengths of SOC were distributed within the following common regions: 400–800, 1380–1440, 1830–1950, and 2090–2400 nm. In these regions, spectral bands were governed by the presence of iron oxides; organic matter; the O–H, C–H, and H–O–H functional groups; kaolin; illite; clay minerals; carbonates; and soil water [46,68,69]. Moreover, these bands are similar to the important wavelength distributions of SOC reported in previous studies, such as those of Hong et al. [68], Jiang et al. [63], Moura-Bueno et al. [49], and Xu et al. [70]. Therefore, when we used EPO to correct the in situ factors of SOC estimation, we could expect a stable distribution of the important wavelengths of SOC.

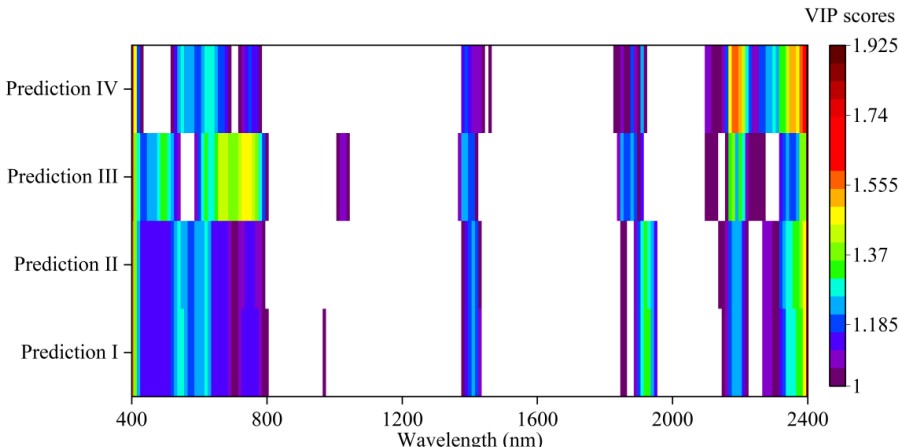

**Figure 10.** Variable importance projection scores extracted from the different PLS models for SOC estimation. The threshold for VIP is 1.

*3.6. Practical Implications*

Most studies rewetted dry ground soils to obtain an EPO transformation matrix and predict SOC contents under different moisture levels [17–19,26]. Few studies have focused on field-moist intact soil prediction, which our study aimed to investigate. In our study, the legacy dry soil samples were wetted to some predefined moisture levels to develop the EPO matrix, which was then applied to field soils. This development process did not involve fieldwork and was relatively simple in practical applications. Conversely, the EPO projection matrix developed from the field data should scan the soil samples under both moist intact and dry ground conditions. However, this method is difficult in practical applications when collecting an adequate number of soil samples for EPO development is impossible because of sampling or economic difficulties [71]. Overall, in our case, in terms of prediction accuracy, *P1* and *P2* slightly differed (Table 2). Our results suggested that EPO *P1* matrix featuring variable moisture levels could minimize sampling requirements and simultaneously reduce the effect of moisture on moist-intact spectra. Ge et al. [15] indicated that their efforts to correct intactness with an EPO algorithm obtains low gain, in contrast to moisture, and variations in natural aggregation on reflectance spectra are limited. Thus, in practical aspects, we could use the *P1* transformation matrix to correct the field-moist intact spectra. The *P1* matrix could be achieved in the laboratory by manually rewetting legacy samples and could effectively simplify the whole modeling process and save time, money, and labor resources. The *P2* matrix should acquire spectral data in moist-intact and dry ground states, thereby increasing sampling time and economic cost, but further improvement in model accuracy is considerable.

**4. Conclusions**

This study investigated the usefulness of the EPO algorithm to reduce the effect of moisture and other in situ factors on field-moist intact soils for SOC prediction. Two types of EPO projection matrices were developed: one from rewetting archive soil samples that had been in a dry ground state in the laboratory (*P1*) and one from the natural state of soils (*P2*). Two different modeling techniques, namely, PLS and MBL, were also compared for SOC estimation. The following conclusions could be drawn from:

(1) The laboratory dry ground soils were less disturbed by external environmental parameters, and accurate estimation models could be achieved for SOC. The RPIQ values of PLS and MBL were 2.70 and 2.91, respectively. Considering the interference of moisture and some other in situ factors, we should have a mathematical algorithm to eliminate the influence of external parameters on the reflectance spectra when a dry soil spectral library would be used to predict moist-intact soils. Without correction, the PLS and MBL had RPIQ of 0.43 and 1.25 for moist-intact soils, respectively.

(2) The performance of the prediction of SOC of field-moist intact soils with a dry spectral library in *P2* was better than that in *P1*. The differences in the validated $R^2$ between the two projection matrices were 0.01 and 0.04 for PLS and MBL, respectively. Nevertheless, the *P1* matrix was recommended for correcting the external parameters because it could effectively simplify the whole modeling process and save time, money, and labor resources.

(3) Local modeling (MBL) performed better than global modeling (PLS) in a large spectral library and could exclude potential sample outliers. With EPO correction, the best SOC prediction for moist intact soils was achieved by the MBL model with *P2* correction, and the validated $R^2$ was 0.69.

(4) The EPO correction did not significantly affect the distribution of the important waveband of SOC. The important wavelengths for SOC estimation were mainly located within 400–800, 1380–1440, 1830–1950, and 2090–2400 nm.

We concluded that it is possible to utilize the EPO correction and local modeling to develop spectral models from dry ground libraries for estimating SOC based on soils collected in the field-moist intact state.

**Author Contributions:** Conceptualization, W.Y.,Y.H. and L.Z.; methodology, W.Y. and Y.C. and S.C.; software, Y.H. and S.C.; validation, L.Z.; formal analysis, W.Y. and Y.H.; investigation, W.Y. and S.C.; resources, Y.C.; data curation, W.Y. and Y.H.; writing—original draft preparation, W.Y.; writing—review and editing, Y.H.; visualization, W.Y. and Y.H.; supervision, L.Z.; project administration, Y.H.; funding acquisition, L.Z. All authors have read and agreed to the published version of the manuscript.

**Funding:** This work was supported by the National Natural Science Foundation of China (grant numbers: 41930754, 32060370) and the Natural Science Foundation of Tibet Autonomous Region (No. XZ 2018ZR G-34(Z)).

**Institutional Review Board Statement:** The study did not involve humans or animals.

**Informed Consent Statement:** The study did not involve humans.

**Conflicts of Interest:** The authors declare no conflict of interest.

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
