# Peer review of "Comparing Two Different Development Methods of External Parameter Orthogonalization for Estimating Organic Carbon from Field-Moist Intact Soils by Reflectance Spectroscopy"

_remotesensing, doi:10.3390/rs14061303_

Round 1

Reviewer 1 Report

This manuscript compares two methods of developing moisture corrections for prediction of SOC on field-moist soil samples. This is an important research topic as the interest in field-deployment of soil spectroscopy continues to grow. It is also a topic that has been fairly well researched in multiple studies over the past decade. This MS, which is generally well written, logically organized and easy to follow, does add some novel insights to this field of research; however, I believe there is a fundamental flaw in the experimental design that really limits the utility of the findings.

A major flaw in the experimental design and perhaps the reason that EPO-P2 out performed EPO-P1 is that the EPO transformation matrix was developed from the same samples that were predicted for “prediction four” while an independent set of samples was used to develop the EPO transformation matrix for “prediction three”. The same exact soil samples should have been used in both cases.

The words “ground” and “milled” are used to describe the dried samples scanned in the laboratory. One of the advantages of VNIR over MIR is that you do not need to finely grind or mill samples prior to scanning. Please describe exactly what you mean and what device was used to grind and mill samples before passing a 2 mm sieve? Typically soils only need to be crushed after drying prior to sieving.

The setup of the ASD Fieldspec is never described. This is critical information especially for the field collection. If a contact probe was used then differences due to solar radiation and atmosphere would have been eliminated in the field. Having to bring the field-moist soils back to the lab really limits the utility of this method. Many other studies have demonstrated the utility of EPO on actual field measured samples.

L121 – there is no such device called ProFR. Did you mean Pro4?

L251 – why was GPL regression chosen? Did you test PLSR and weighted PLSR?

L490-8 – I want to believe these statements, and it is very possible that they are true, but with the transfer matrices being generated on different samples it is really hard to come to these conclusions from the data presented in this paper. Additionally, if these statements are true, then wouldn’t the logical conclusion of the entire paper be that P2 is superior to P1 even if it involves slightly more cost/effort.

Fig 9 and 10 – these color ramps are very hard to distinguish low and high values especially in B&W. There are lots of great resources for picking good sequential color gradients such as ColorBrewer (https://colorbrewer2.org/#type=sequential&scheme=BuGn&n=3).

I don’t really agree with the practical implications section. Rewetting and slowly drying a 120 samples and scanning them multiple times over the course of several weeks is a lot of labor. For P2, the only real increase in effort was the need to dry and rescan the samples since the samples needed to be collected and brought back to the lab for the field-moist scans for both prediction three and prediction four.

Reviewer 2 Report

The concluding paragraph needs to refer back to the final sentence of he abstract. It is important for the reader to understand whether the research was for economic reasons or a better estimation of the amount of SOC or possibly both.

Reviewer 3 Report

The manuscript entitled “Comparing two different development methods of external parameter orthogonalization for estimating organic carbon from field-moist intact soils by reflectance spectroscopy” demonstrates that (1) external parameter orthogonalization (EOP) is effective for modelling soil organic carbon (SOC), and (2) the locally based memory-based learning (MBL) method is better than the globally based partial least square (PLR) method in modelling SOC from reflectance spectroscopy. Although using field-moist intact data gains a better accuracy, the laboratory-rewetting data are also acceptable. The manuscript is well written and easy to follow.

Reviewer 4 Report

This paper introduces a decomposition method of field measured spectral curve, which retains effective information and removes interference information through EPO algorithm, so as to improve the accuracy of SOC estimation. This work is very meaningful. The feasibility of the EPO method is verified by abundant data. Nevertheless, there is still a problem: why can EPO algorithm effectively separate interference information and what is its principle?

Reviewer 5 Report

The manuscript is well-written and investigates two methods for correcting for soil moisture while predicting SOC. Although the use of the EPO correction is not new, the authors tested the effect of either re-wetting grounded dry soils from their spectral library or using fresh soil collected in the field. The latter method performed slightly better, but the authors conclude that the benefits do not weigh up to the extra costs of collecting fresh samples. The study could have some important implications for predicting SOC from airborne or remote sensing platforms where the soil moisture content in the pixels is difficult to quantify. Some sentences in the discussion could be devoted to this issue.

Line 30 What does respectively mean in this sentence: Data set A for calibration and Dataset C for validation? If so, to state:’ Dataset A was used for calibration the moist intact samples in Dataset C for validation’

Line 45 and throughout the manuscript Please consider changing ‘concentration’ in ‘content’. Concentration refers to dissolution in a liquid, while content is more general and can also apply to solids (as is the case with SOC).

Line 27 From the M&M section I understand that these 120 samples are ‘dataset B’.

Line 303 I would suggest to provide the moisture content with one decimal number.

Line 308 Please indicate on which axis the variables are plotted in brackets. What do the error bars represent?
